# *Lactiplantibacillus plantarum* for the Preparation of Fermented Low-Bitter Enzymatic Skim Milk with Antioxidant Ability

**DOI:** 10.3390/foods13233828

**Published:** 2024-11-27

**Authors:** Yi Jiang, Longfei Zhang, Yushi Jin, Haiyan Xu, Yating Liang, Zihan Xia, Chenchen Zhang, Chengran Guan, Hengxian Qu, Yunchao Wa, Wenqiong Wang, Yujun Huang, Ruixia Gu, Dawei Chen

**Affiliations:** 1College of Food Science and Engineering, Yangzhou University, Yangzhou 225127, China; jy15098598780@163.com (Y.J.); zlfjssz@126.com (L.Z.); 19818073766@163.com (Y.J.); 19852318085@163.com (H.X.); tiya598@163.com (Y.L.); zihxia@jou.edu.cn (Z.X.); cczhang@yzu.edu.cn (C.Z.); 006456@yzu.edu.cn (C.G.); quhengxian@163.com (H.Q.); 008213@yzu.edu.cn (Y.W.); 006915@yzu.edu.cn (W.W.); yjhuang@yzu.edu.cn (Y.H.); guruixia1963@163.com (R.G.); 2Jiangsu Provincial Key Laboratory for Probotics and Dairy Deep Processing, Yangzhou 225127, China

**Keywords:** Protamex, enzymatic skim milk, *Lactiplantibacillus plantarum* 16, bitternes, antioxidant ability

## Abstract

A high degree of hydrolysis can reduce the allergenicity of milk, while lactic acid bacteria (LAB) fermentation can further enhance the antioxidant ability of enzymatic milk. LAB with a strong antioxidant ability was screened, and the effects of LAB on the bitterness, taste and flavor of enzymatic skim milk (ESM) with a high degree of hydrolysis were investigated in this paper, in addition to the response surface methodology optimized the conditions of the LAB fermentation of ESM. The results indicate that the skim milk hydrolyzed by Protamex has a higher degree of hydrolysis and lower bitterness. The scavenging rate of 2,2-Diphenyl-1-picrylhydrazyl (DPPH) free radical, the inhibition rate of hydroxyl radical (·OH) and the superoxide dismutase (SOD) activity of *Lactiplantibacillus plantarum* 16 and *Lactococcus lactis* subsp. *lactis* m16 are significantly higher than those of other strains (*p* < 0.05), while the improvement effect of *L. plantarum* 16 on the bitterness and flavor of ESM is better than that of *L. lactis* subsp. *lactis* m16. The fermented ESM has a strong antioxidant ability and low bitterness when the inoculum quantity of *L. plantarum* 16 is 5%, fermentation at 37 °C for 18 h and the pH of the ESM is 6.5, for which the DPPH free radical scavenging rate is 61.32%, the ·OH inhibition rate is 83.35%, the SOD activity rate is 14.58 and the sensory evaluation is 4.25. The contents of amino acids related to bitterness and antioxidants were reduced and increased, respectively. The ESM fermented by *L. plantarum* 16 has a good flavor, antioxidant ability and low bitterness.

## 1. Introduction

Milk serves as a superior source of protein for the human body, which constitutes a vital component of tissues such as muscles and bones, furnishing essential amino acids necessary for nourishing the body, participating in the modulation of multiple physiological functions, and acting as a significant source of energy for the organism. Simultaneously, the nutritious constituents like minerals, vitamins and immune-active substances it contains exert a positive stimulative effect on the body’s health. Nevertheless, owing to specific disparities with human milk protein [1], it frequently gives rise to allergic symptoms such as skin eczema and rhinitis [2]. In recent years, on account of the high desensitization efficiency, mild reaction and environmentally friendly attributes of protease, it has been extensively employed in reducing the allergenicity of food. It achieves this by disrupting the intramolecular and intermolecular chemical bonds within milk proteins, generating low-molecular-weight peptides or amino acids to modify the protein conformation and linear epitopes [3]. Not only can it attain the outcome of reducing the allergenicity of milk [4], but it can also diminish the damage caused by hydroperoxides and reactive oxygen in the body [5]. Skim milk is also more conducive to enzymatic hydrolysis [6]; therefore, enzymatic skim milk (ESM) benefits more people regarding milk’s nutrients and, meanwhile, makes them more easily absorbed by the human body and enhances their antioxidant and other functional properties [7,8].

The degree of hydrolysis is a crucial indicator for gauging the extent of the enzymatic hydrolysis of proteins. ESM with a high degree of hydrolysis can not only enhance the body’s antioxidant ability by scavenging 2,2-Diphenyl-1-picrylhydrazyl (DPPH) free radical and inhibiting hydroxyl radical (·OH) [7,8], but also reduce the allergenicity of skim milk and exert significant influences on its emulsification, gelation and other characteristics [9]. Nevertheless, a relatively high degree of hydrolysis will increase the content of bitter substances, such as bitter peptides and amino acids, in the enzymatic skim milk [10,11], thereby restricting its application in food. Both physical–chemical and biological approaches can be employed to reduce the bitterness of skim milk. However, physical–chemical methods such as bitterness masking and the Maillard reaction can disrupt the protein structure and inactivate bioactive peptides. However, biological debittering methods such as proteolytic hydrolysis possess advantages like mild conditions and stable effects [12]. In recent years, growing evidence shows that flavoenzyme, protamex and neutrons are widely used in the enzymatic hydrolysis of milk due to their ability to enhance the solubility, apparent viscosity and wettability of milk protein while reducing the production of bitterness [13,14,15].

With the deepening of research, people have found that lactic acid bacteria (LAB) play a crucial role in human health; therefore, they are increasingly being used in food to enhance their antioxidant and other probiotic functions. However, the impact of LAB with a solid antioxidant capacity on the bitterness and flavor of food is still unclear.

Consequently, with the degree of hydrolysis as the evaluation index, a protease with a stronger ability to hydrolyze skim milk was screened out from flavorzyme, protamex and neutrose in this paper. Then, the ESM was taken as the fermentation substrate and LAB with a strong DPPH free radical scavenging rate and ·OH inhibition rate, as well as high superoxide dismutase (SOD) activity, were screened out. The effect of LAB with a potent antioxidant ability on improving the bitterness and flavor of ESM was further investigated. The response surface methodology further optimized the conditions of LAB fermenting ESM and ESM with lower bitterness and a more vital antioxidant ability was prepared, which provided a reference basis for developing a fermented milk beverage with antioxidant activity, a low level of bitterness and related probiotic products.

## 2. Materials and Methods

### 2.1. Bacterial Strains and Growth Conditions

*Lactiplantibacillus plantarum* (16, 662, 13, 31, 144), *Limosilactobacillus fermentum* (69, 57, 43M-2, 630), *Lacticaseibacillus rhamnosus* m98, *Lacticaseibacillus paracasei* M111, *Lactococcus lactis* subsp. *Lactis* m16 and *Bifidobacterium breve* (S10, W1P2) were isolated and preserved by the Jiangsu Provincial Key Laboratory for Probiotics and Dairy Deep Processing. They were cultured at 37 °C in MRS broth, M17 broth and Modified MRS broth (Hope Bio-Technology Co., Ltd., Qingdao, China) until the stationary phase was reached.

### 2.2. Preparation of ESM

A 12% (*w*/*w*) skim milk (Skim Milk Powder, Fonterra Co-operative Group Ltd., Auckland, New Zealand) solution was prepared using pure water, which was hydrolyzed by Neutrase, Flavorzyme and Protamex (Novozymes Biotechnology Co., Ltd., Tianjin, China) following the manufacturers instruction, respectively. Samples were taken after enzymatic hydrolysis for 4.5 h, placed in an autoclave at 105 °C for 5 min to deactivate the enzymes, and then stored in a refrigerator at 4 °C for subsequent study. The enzymatic hydrolysis conditions are presented in Table 1.

### 2.3. Determination of the Hydrolysis Degree of ESM

The pH-Stat method was used to determine the degree of hydrolysis of the ESM, according to Guowei et al. [16]. The 0.1 mol/L NaOH (food grade; Shandong Binhua Dongrui Chemical Co., Ltd., Qingdao, China) solution was dropped into the reactants regularly to maintain a constant pH value during the enzymatic reaction process, and the volume of consumed NaOH was recorded. The degree of hydrolysis was calculated following Guowei et al. [16].

### 2.4. Sensory Evaluation of the Bitterness of ESM

Standard solutions of quinine were prepared with concentrations of 5 × 10^−5^, 1 × 10^−4^, 2 × 10^−4^, 4 × 10^−4^, 8 × 10^−4^ and 1.6 × 10^−3^ g/mL, corresponding to scores of 0, 2, 4, 6, 8 and 10 points, with lower bitterness resulting in lower scores. Ten graduate students in food science participated in the evaluation, including five men and five women. The evaluators compared the samples with the standard solutions at room temperature and scored the test samples according to Huang et al. [17].

### 2.5. Determination of the Bitterness Response Value of ESM

The bitterness of ESM was measured by the INSERT SA402B electronic tongue (Insert Intelligent Sensor Technology Co., Ltd., Kanagawa, Japan). The sensor consisted of four sensors, including sourness, bitterness, saltiness and umami, which were rinsed with fresh reference solution for 6 s and then immersed again in the reference solution. A cleaning solution was washed on the electrode for 90 s and a reference solution was washed for 180 s before testing. The operation procedure follows Fagundes et al. [18].

### 2.6. Preparation of LAB-Fermented ESM

LAB was centrifuged at 8000× *g* for 10 min to collect the bacterial cells, which were washed with sterile water twice and the concentration was adjusted to 1 × 10^9^ CFU/mL. Then, they were then inoculated into enzyme-hydrolyzed skim milk at a 3% inoculum quantity at 37 °C for 18 h and stored at 4 °C in the refrigerator for future study.

### 2.7. Determination of Antioxidant Ability

#### 2.7.1. Determination of the Scavenging Ability of DPPH Free Radical

The samples were added to the DPPH reagent (Nanjing Jiancheng Biological Engineering Company, Nanjing, China) and the reaction mixture was thoroughly mixed by shaking and then incubated in the dark for 30 min at 25 °C. After that, the mixture was taken to measure the absorbance values at 517 nm against a blank. The detailed methods of DPPH free radical scavenging rate of LAB followed the method of Zhang et al. [19].

#### 2.7.2. Determination of the Inhibitory Ability Against ·OH

The samples were mixed with ·OH reagent (Nanjing Jiancheng Biological Engineering Company, Nanjing, China) after diluting 150 times with the pure water and then incubated at 37 °C in a water bath. The absorbance was determined at 550 nm with the blank comprising reagents. Hao et al. described the determination method of inhibitory ability against ·OH of LAB [20].

#### 2.7.3. The Determination of SOD Activity

The samples were added to the SOD reagent (Nanjing Jiancheng Biological Engineering Company, Nanjing, China) and the reaction mixture was thoroughly mixed by shaking and then incubated at 37 °C for 20 min. After that, the mixture was taken to measure the absorbance values at 450 nm against a blank. The determination method of the SOD activity of LAB was described by Hucheng et al. [21].

### 2.8. Determination of Volatile Flavor Compounds in ESM Fermented by LAB

Solid-phase microextraction (SPME) fiber (Supelco, Inc., Bellefunte, PE, USA) used for the first time was inserted into the injection of the Agilent 7890B gas chromatograph (Agilent Technologies, Inc., Santa Clara City, CA, USA) at 250 °C until no chromatographic peaks emerged. The baseline was stable and then the SPME fibers were inserted into the gas-phase bottle that contained a sample for extraction for 60 min. Desorption occurred at 250 °C for 3 min while the instrument started to collect data.

A temperature-programmed method was used for chromatography. The DB-WAX column (30 m × 0.25 mm, 0.25 μm; Agilent Technologies, Inc., Santa Clara City, CA, USA) was initially held at 35 °C, then increased by 4 °C/min to 140 °C and, finally, increased to 250 °C for 3 min. The transfer line temperature was set to 250 °C. Helium was used as the carrier gas with a 1 mL/min flow rate and no split sampling was performed.

For mass spectrometry, electron ionization was performed at 70 eV, ion source temperature at 230 °C, mass scan range (*m*/*z*) was 33–450 AMU and emission current was 100 μA.

The operations of qualitative analysis and evaluation of odor activity are provided by Dan et al. [22].

### 2.9. Optimization of the Fermentation Conditions for the Fermentation of ESM by LAB

#### 2.9.1. Single-Factor Experiment on Fermentation of ESM by LAB

LAB, with strong antioxidant ability and sound improvement effects on ESM’s bitterness and flavor, was used to ferment ESM. The impact of fermentation temperature, fermentation time, inoculum quantity and pH of ESM on antioxidant ability was investigated and the optimal fermentation conditions range was confirmed. The specific conditions of the single-factor experiment are as follows:(1)The fermentation time is 18.0 h, the inoculum quantity of LAB is 3.0% and the pH of ESM is 6.5. The fermentation temperatures are 25.0, 29.0, 33.0, 37.0, 41.0 and 45.0 °C, respectively.(2)The fermentation temperature is 37.0 °C, the inoculum quantity of LAB is 3.0% and the pH of ESM is 6.5. The fermentation times are 10.0, 12.0, 14.0, 16.0, 18.0 and 22.0 h, respectively.(3)The fermentation temperature is 37.0 °C, the fermentation time is 18.0 h and the pH of ESM is 6.5. The inoculum quantities of LAB are 1.0%, 2.0%, 3.0%, 4.0%, 5.0% and 6.0%, respectively.(4)The fermentation temperature is 37.0 °C, the fermentation time is 18.0 h and the inoculum quantity of LAB is 3.0%. The pH values of ESM are 4.5, 5.0, 5.5, 6.0, 6.5 and 7.0, respectively.

#### 2.9.2. Response Surface Experiment on the Fermentation of ESM by LAB

The appropriate level ranges of each factor were selected. Taking the LAB inoculum quantity, fermentation temperature, fermentation time and substrate pH as 4 independent variables and the fuzzy comprehensive evaluation value with the bitterness evaluation value of LAB-fermented ESM, DPPH free radical scavenging rate, ·OH inhibition rate and SOD activity as response values, the fermentation conditions of LAB-fermented ESM were optimized through the response surface experiment. The factor level design of the response surface experiment is shown in Table 2 and the data were analyzed.

### 2.10. The Determination of Amino Acid Content

The hydrolysis tube containing fermented ESM was added to 6 mol/L HCl solution. After adding 3–4 drops of phenol, which was placed in the refrigerator for 3 to 5 min and evacuated to a near vacuum (close to 0 Pa), the following operations were carried out according to Jaudzems et al. [23].

### 2.11. Statistics and Analysis

GraphPad Prism 10 (GraphPad Software Inc.; San Diego, CA, USA) was used to analyze all data expressed as means ± SD. One-way ANOVA was used to compare group differences, followed by Tukey’s post hoc multiple comparison test. Data are considered statistically different from each other at *p* < 0.05.

## 3. Results

### 3.1. The Impact of Protease on the Degree of Hydrolysis of ESM

It can be known from Figure 1 that the hydrolysis degree of ESM by the Protamex is 28.13%, which is significantly higher than 24.89% of the Neutrase and 23.20% of the Flavorzyme (*p* < 0.05), suggesting that the degree of hydrolysis of skim milk by the Protamex is relatively high.

### 3.2. The Effect of Proteases on the Bitterness of Skim Milk

#### 3.2.1. The Sensory Evaluation of the Bitterness of ESM

It can be seen from Figure 2 that the bitterness evaluation value of ESM by Neutrase was 7.60, which was significantly higher than 6.80 of the Protamex and 6.60 of the Flavorzyme (*p* < 0.05), suggesting that the bitterness of skim milk enzymatically hydrolyzed by Flavorzyme and Protamex was lower.

#### 3.2.2. The Bitterness Response Value of ESM

As can be observed from Figure 3, quinine’s bitter taste response value gradually increases with the increase in its concentration, and the concentration of the flavor substance and intensity exhibits an S-shaped pattern. Within the range of the quinine concentration at 4 × 10^−4^ g/mL, the growth rate of the bitter taste response value is higher than that of the concentration, which is the “expansion stage”. When the quinine concentration is between 4 × 10^−4^ and 8 × 10^−4^ g/mL, there is a linear relationship between the bitter taste response value and the concentration. When the quinine concentration exceeds 8 × 10^−4^ g/mL, the growth of the bitter taste response value tends to stabilize and no longer undergoes significant changes. The relationship between the response value and the concentration at this stage can be fitted by the logarithmic function y = 3.84 × ln(x) + 31.55 (R^2^ = 0.9212). We can calculate the quinine equivalent value corresponding to the sample within a specific range of bitter taste response values with relatively high accuracy through this function.

The bitter taste results in Figure 4 indicate that the bitter response value of the Protamex was 5.73, which was lower than 5.97 of the Flavorzyme and 6.23 of the Neutrase.

The bitter response values of the Neutrase, the Flavorzyme and the Protamex are between 5–7 (Figure 4) and at this time, the quinine concentration is 4 × 10^−4^–8 × 10^−4^ g/mL (Figure 3), corresponding to a bitter value within the range of 6–8 points (Figure 2). This indicates that the sensory evaluation is relatively consistent with the results of the electronic tongue, demonstrating that the combined application of artificial sensory evaluation and intelligent sensory evaluation methods in the bitter ESM assessment is reliable.

It is discovered that the skim milk enzymatically hydrolyzed by the Protamex has a lower bitterness according to comprehensively considering the hydrolysis degree of ESM, the bitter sensory evaluation and the electronic tongue taste results. Therefore, the subsequent fermentation study used the enzymolysis of skim milk by Protamex.

### 3.3. The Antioxidant Ability of LAB

Figure 5 shows that the DPPH free radical scavenging rate of *L. plantarum* 16 was 57.91%, significantly higher than that of other strains (*p* < 0.05); the ·OH inhibition rate of *L. plantarum* 16 and *L. lactis* subsp. *lactis* m16 were significantly higher than those of other strains (*p* < 0.05), being 81.58 and 81.72%, respectively, and their SOD activity was also significantly higher than those of different strains (*p* < 0.05), being 14.27 and 14.48 U/mL, respectively. This indicates that *L. plantarum* 16 and *L. lactis* subsp. *lactis* m16 can better enhance the antioxidant ability of the ESM of the Protamex.

### 3.4. The Effect of LAB Fermentation on the Bitterness of ESM

#### 3.4.1. The Evaluation Value of Bitterness of Fermented ESM

It can be known from Figure 6 that the bitterness evaluation value of ESM fermented by *L. plantarum* 16 was 4.80, which was significantly lower than 6.15 of *L. lactis* subsp. *lactis* m16 (*p* < 0.05). Although the bitterness evaluation value of ESM fermented by *L. plantarum* 16 was not significantly lower than that of ESM (*p* > 0.05), it still reduced the bitterness of ESM to a certain extent. On the contrary, the fermentation of *L. lactis* subsp. *lactis* m16 significantly increased the bitterness evaluation value of ESM (*p* < 0.05).

#### 3.4.2. The Response Value of Bitterness of Fermented ESM

The bitterness response value of the ESM fermented by *L. plantarum* 16 was 5.10, which was lower than 7.77 of *L. lactis* subsp. *lactis* m16, and also significantly lower than 5.73 of the unfermented milk (Figure 7). Meanwhile, the astringency response value of the ESM fermented by *L. lactis* subsp. *lactis* m16 was significantly higher than that of *L. plantarum* 16 (*p* < 0.05) and the astringency had an essential influence on the taste of the ESM. This indicates that *L. plantarum* 16 can significantly improve the bitterness of the ESM while the fermentation of *L. lactis* subsp. *lactis* m16 enhances the bitterness of the ESM.

### 3.5. The Effect of LAB Fermentation on the Volatile Flavor Substances in ESM

It can be known from Table 3 that the volatile flavor components in the fermented and unfermented ESM were mainly composed of aldehydes, alcohols, ketones, esters, acids and heterocyclic compounds. In the ESM, the aldehyde substances with relatively high contents are benzaldehyde, 2,4-dimethylbenzaldehyde and natural nonanal, with the relative contents being 5.68, 1.29 and 1.07%, respectively. The alcohol substance with a relatively high content is octanol, which is 1.34%. Among the ketone substances, those with relatively high contents are 2-decanone, 2-nonanone and 2-heptanone, with relative contents of 1.19, 0.34 and 0.30%, respectively.

In the ESM fermented by *L. lactis* subsp. *lactis* m16, the aldehyde substances with relatively high contents are 2,4-dimethylbenzaldehyde, benzaldehyde and natural nonanal, with the relative contents being 2.96, 2.59 and 0.35%, respectively. The alcohol substances with relatively high contents are octanol, nonanol and 1-heptanol, with relative contents of 3.11, 1.92 and 0.63%, respectively. The ketone substances with relatively high contents are isophorone, 2-heptanone and 2-nonanone, with relative contents of 3.39, 0.61 and 0.49%, respectively.

### 3.6. Optimization of the Fermentation Conditions of ESM by L. plantarum 16

#### 3.6.1. Single-Factor Experiment

##### The Effect of Fermentation Temperature on the Antioxidant Ability of ESM Fermented by *L. plantarum* 16

It can be found that *L. plantarum* 16 presents a strong antioxidant ability and sound improvement effects on the bitterness and flavor of ESM; therefore, *L. plantarum* 16 was screened out to ferment the ESM (Protamex enzymolysis) and the fermentation conditions were optimized.

As can be observed from Figure 8, the fermentation temperature has a significant influence on the antioxidant ability of ESM during fermentation (*p* < 0.05). The DPPH free radical scavenging rate, ·OH inhibition rate and SOD activity all exhibit a trend of initially increasing and then decreasing with the increase in the fermentation temperature. Specifically, the DPPH free radical scavenging rate reaches its maximum at a fermentation temperature of 37.0 °C, amounting to 57.00%, followed by 33.0 and 41.0 °C, with scavenging rates of 53.86 and 52.36%, respectively. The ·OH inhibition rate attains its peak at a fermentation temperature of 33.0 °C, reaching 85.62%, followed by 29 °C and 37 °C, with inhibition rates of 77.94 and 81.65%, respectively. The SOD activity also reaches its maximum at 33.0 °C, followed by 29 and 37 °C. Considering the results of the comprehensive antioxidant ability and further detailing the effect of the temperature on the antioxidant ability, the fermentation temperatures of 33.0, 35.0 and 37.0 °C were selected for subsequent response surface experiments.

##### The Effect of Fermentation Time on the Antioxidant Ability of ESM Fermented by *L. plantarum* 16

It can be seen from Figure 9 that the fermentation time has a significant influence on the antioxidant ability of the ESM during fermentation (*p* < 0.05). The DPPH free radical scavenging rate, ·OH inhibition rate and SOD activity all show a trend of initially increasing and then decreasing with the extension of the fermentation time. Specifically, the DPPH free radical scavenging rate reaches its maximum of 59.10% at a fermentation time of 16.0 h, followed by 14 and 18 h, with the scavenging rates being 57.22 and 57.74%, respectively. The ·OH inhibition rate reaches its peak of 84.59% at a fermentation time of 16 h, followed by 14 and 18 h. The SOD activity reaches its maximum at a fermentation time of 16 h, followed by 14 and 18 h. Based on the comprehensive antioxidant ability results, fermentation times of 14.0, 16.0 and 18.0 h were selected for the subsequent response surface experiments.

##### The Effect of Inoculum Quantity on the Antioxidant Ability of *L. plantarum* 16-Fermented ESM

It can be seen from Figure 10 that the inoculum quantity of *L. plantarum* 16 has a significant influence on the antioxidant ability of the ESM during fermentation (*p* < 0.05). The DPPH free radical scavenging rate reaches its maximum of 64.26% when the inoculum quantity is 4%, followed by 3 and 5%. The ·OH inhibition rate attains its peak of 82.17% when the inoculum quantity is 4%, followed by 3 and 5%, with the inhibition rates being 80.35 and 80.14%, respectively. The SOD activity reaches its highest value when the inoculum quantity is 3%, followed by 2 and 4%. Considering the comprehensive antioxidant ability results, the inoculum quantities of 3.0, 4.0 and 5.0% were selected for the subsequent response surface experiments.

##### The Effect of ESM pH on the Antioxidant Ability of *L. plantarum* 16 Fermentation

As can be seen from Figure 11, the pH of ESM has a significant influence on the antioxidant ability of fermented ESM (*p* < 0.05). The DPPH free radical scavenging rate reaches the maximum of 60.22% at pH 6.0 of ESM, followed by pH 5.5 and pH 6.5, with the scavenging rates being 59.42 and 57.55%, respectively. The ·OH inhibition rate reaches the maximum of 84.88% at pH 5.5 of ESM, followed by pH 5.0 and pH 6.0, with the inhibition rates being 83.55 and 83.41%, respectively. The SOD activity reaches the maximum at pH 6.5 of ESM, followed by pH 6.0 and pH 7.0. Considering the comprehensive antioxidant ability results, the pH of the ESM was selected as 5.5, 6.0 and 6.5 for the subsequent response surface experiments.

#### 3.6.2. The Results of the Response Surface Experiment

##### Response Surface Optimization Scheme and Results

A response surface optimization was conducted with fermentation temperatures, fermentation time, inoculum quantity, and ESM pH as the four independent variables, and the fuzzy comprehensive evaluation value was considered as the response value according to the Box-Behnken central composite design principle; the experimental scheme and results are shown in Table 4. 

The data analysis was conducted using Design-Expert 11 software (Stat-Ease, Inc., Minneapolis, MN, USA) and the following quadratic polynomial regression equation was obtained: Y = 0.5009 + 0.0761A + 0.0193B − 0.0075C + 0.0263D − 0.0502AB − 0.0241AC + 0.0505AD + 0.0141BC + 0.0250BD − 0.0224CD + 0.0579A^2 − 0.0181B^2 − 0.0436C^2 + 0.0307D^2. In the equation, A represents the fermentation time, B represents the fermentation temperature, C represents the inoculum quantity, D represents the pH of ESM and Y represents the fuzzy comprehensive evaluation value.

It can be known from Table 5 that the F value of this model is 12.75 (*p* < 0.05), which is statistically significant, and the lack-of-fit term is not substantial (*p* = 0.1665 > 0.05), demonstrating that this mathematical model is appropriate. The determination coefficient R^2^ = 0.9273 and the adjusted determination coefficient R^2^ Adj = 0.8545 suggest that the equation has an excellent fitting degree and it is feasible to predict and determine the optimal process conditions using this model. According to the magnitude of the *p* value, the contribution rate of each factor to the response value can be reflected as A > D > B > C, that is, fermentation temperature > ESM > fermentation time > pH > inoculum quantity.

##### Response Surface Interaction and Contour Graph

Figure 12 reveals the impact of the fermentation time and inoculum quantity in the interaction with the fuzzy comprehensive evaluation value of *L. plantarum* 16 in the fermentation enzymolysis process. By observing the slope changes of the response surface, it is possible to intuitively evaluate the degree of influence of different factors on the fuzzy comprehensive evaluation value. The steeper the slope, the more significant the factor’s influence on the evaluation value. The shape of the contour lines can reflect the strength of the interaction, with elliptical contour lines indicating a significant interaction and shapes closer to circular indicating a relatively weak interaction.

It can be known from Figure 9 that there is a noticeable slope change between the fermentation time and the inoculum quantity and the contour shape is approximately elliptical. It reflects that the interaction between the two significantly influences the fuzzy comprehensive evaluation value of *L. plantarum* 16 in the process of enzymatic hydrolysis and the defatting of milk (*p*AB < 0.05).

##### More in Line with the Academic Context

Considering the actual situation, the optimal fermentation conditions were obtained as a fermentation temperature of 37 °C, a fermentation time of 18 h, an inoculum quantity of 5%, a pH of 6.5 for ESM and a predicted fuzzy comprehensive evaluation value of 0.74. Under these conditions, the sensory evaluation of enzymatically hydrolyzed skim milk fermented by *L. plantarum* 16 was measured as 4.25, the DPPH free radical scavenging rate was 61.32%, the ·OH inhibition rate was 83.35% and the SOD activity rate was 14.58.

### 3.7. The Content of Amino Acids in ESM Fermented by L. plantarum 16

It is evident from Table 6 that, in comparison with the unfermented ESM, the fermentation of *L. plantarum* 16 did not significantly alter the total amount of amino acids in the ESM (*p* > 0.05); however, it significantly increased the contents of glutamic acid, glycine, alanine, methionine, lysine and arginine in the ESM (*p* < 0.05). Meanwhile, it significantly decreased the contents of serine, tyrosine, phenylalanine and histidine in the ESM (*p* < 0.05). It may suggest that *L. plantarum* 16 fermentation significantly changed the amino acid content of ESM, which is closely related to the bitter taste and antioxidant capacity.

## 4. Discussion

Enzymolysis can decompose macromolecular proteins into small-molecule peptides or amino acids. The more peptide bonds of broken proteins, the higher the degree of hydrolysis. This indicates a complete hydrolysis of the protein and confers properties such as desensitization and the antioxidant ability to the milk [14,24]. With the increase in hydrolysis, the protein peptide chains hydrolyzed by proteases gradually unfold, generating more bitter peptides [25]. Hence, bitter substances often accompany the enzymatic hydrolysis of proteins. In contrast, the Protamex contains broad-spectrum endopeptidases that can extensively hydrolyze proteins and effectively remove hydrophobic amino acids from the peptide termini, completely degrading bitter peptides into amino acids and small peptides, thereby reducing the bitterness of the enzymatic hydrolysate [26]. Thus, the Protamex is capable of lowering the bitterness of ESM.

Our research discovered that the bitterness of ESM fermented by *L. plantarum* 16 was significantly lower than that of *L. lactis* subsp. *lactis* m16 (*p* < 0.05). Possibly, compared to *L. plantarum* 16, *Lactococcus lactis* is more likely to produce endopeptidases during metabolism and the increase in the endopeptidase content causes the exposure of hydrophobic amino acids at the termini of peptides, further augmenting the bitterness of ESM [27,28]. Through an electronic tongue analysis, it was also found that the umami taste of ESM fermented by *L. plantarum* 16 was higher than that of *L. lactis* subsp. *lactis* m16, while the bitterness was lower (Figure 7). It might be that *L. plantarum* 16 is more capable of converting bitter amino acids into umami-tasting acetyl derivatives [29], thereby reducing the bitterness of ESM fermented by *L. plantarum* 16. The bitter taste evaluation value of the fermented ESM after response surface optimization was 4.25, significantly lower than the unoptimized 6.80, indicating that the optimized fermentation conditions of *L. plantarum* 16 can better reduce the bitter taste property of ESM. Moreover, in Table 6, we further discovered that *L. plantarum* 16 significantly decreased the contents of tyrosine, phenylalanine and histidine in ESM (Table 6) and these amino acids contain functional groups such as carbonyl or amino groups, which release bitter substances under the action of gastric acid [28,30]. Therefore, reducing the content of bitter amino acids is an important approach for *L. plantarum* 16 fermentation to lessen the bitterness of ESM.

LAB fermentation has a positive effect on aspects such as the flavor and functional activity of milk [31]. This study found that there were majority of aldehyde substances such as heptanal, natural nonanal and benzaldehyde, as well as a small quantity of alcohol substances such as octanol and alcohols in the ESM. While aldehyde substances contribute significantly to the fishy smell and generate unpleasant odors, other substances contribute other odors, such as benzaldehyde with almond and wood flavors, nonanal with a butter flavor and decanal with a wax flavor; alcohol and ketone substances contribute more to the fragrance, such as octanol with sweet and grassy flavors and isopentanol with a green apple flavor [32]. The fermentation of *L. plantarum* 16 reduces the aldehyde substances in the ESM from 8.53% to 1.27%, effectively lowering the relative content of aldehyde substances; simultaneously, it effectively increases the relative content of alcohol substances such as isopentanol, 1-decanol and octanol from 1.34% to 13.00%. It might be that *L. plantarum* 16 catalyzes the redox transformation of aldehyde substances in the ESM, converting the aldehyde substances with undesirable flavors into alcohol substances with favorable flavors and ketone substances that can enrich the product flavor [33]. This study also discovered that 2-hydroxy-4-methylpentanoic acid was not detected in both the unfermented ESM and the ESM fermented by *L. lactis* subsp. *lactis* m16, while the relative content in the ESM fermented by *L. plantarum* 16 was 0.28%. 2-Hydroxy-4-methylpentanoic acid not only possesses a good fruit flavor [34], but also can improve inflammation, fibrosis and metabolic disorders in the organism [35], indicating that the fermentation of *L. plantarum* 16 is not only conducive to the improvement of the flavor of ESM, but also has probiotic functions for the organism.

It was found that the antioxidant capacity of *L. plantarum* 16-fermented ESM gradually increased with the extension of the fermentation time in our study. The antioxidant capacity at 18 h of fermentation was higher than that at 14 and 16 h, which may be due to active substances related to the antioxidant activity increasing as the fermentation proceeded [36]. We found that the temperature was another factor that can influence the antioxidant capacity of ESM, and the antioxidant capacity at 37 °C of *L. plantarum* 16-fermented ESM was higher than that of other temperatures, which might suggest that an appropriate temperature can increase the content of antioxidants substances produced by bacterial metabolism [37]. A proper pH of the fermentation substrate not only facilitates the growth of the strain, but also facilitates the dissolution of antioxidant active substances and then improves the antioxidant ability of the fermentation solution, such as scavenging DPPH free radical [38]. This may be an essential factor in the fact that pH 6.5 is more conducive to improving the antioxidant capacity of *L. plantarum* 16-fermented ESM when compared with other pH values.

Our study also found that the fermentation of *L. plantarum* 16 significantly increased the contents of glutamic acid, glycine, alanine, methionine, lysine and arginine in ESM (*p* < 0.05). These amino acids have strong abilities to scavenge DPPH free radicals and inhibit ·OH and enhance SOD activity [39,40,41].

Moreover, glutamic acid and glycine are essential amino acids for synthesizing glutathione [42]. Simultaneously, the extracellular polysaccharides metabolized by *L. plantarum* during the fermentation process also enhanced its ability to scavenge DPPH free radicals [43]. Therefore, the optimized fermentation time, fermentation temperature and fermentation substrate pH of *L. plantarum* 16 may enhance the antioxidant capacity of fermented ESM by increasing the contents of amino acids and related metabolites with antioxidant functions.

## 5. Conclusions

The enzymatic hydrolysis of skim milk with the Protamex has a high degree of hydrolysis and low bitterness. The fermentation of *L. plantarum* 16 can improve the flavor of the ESM by reducing the relative content of aldehyde substances and increasing the relative contents of alcohol and ketone substances. Meanwhile, it can also exert its better effect of reducing the bitterness of the ESM and enhancing its antioxidant ability by regulating the contents of related amino acids. When the inoculum quantity of *L. plantarum* 16 is 5%, the fermentation temperature is 37 °C for 18 h and the pH of the ESM is 6.5, the fermented ESM has a strong antioxidant ability and a low bitterness. This study provides a theoretical basis for developing antioxidant, low-bitterness live-bacteria-fermented dairy beverages.

## Figures and Tables

**Figure 1 foods-13-03828-f001:**
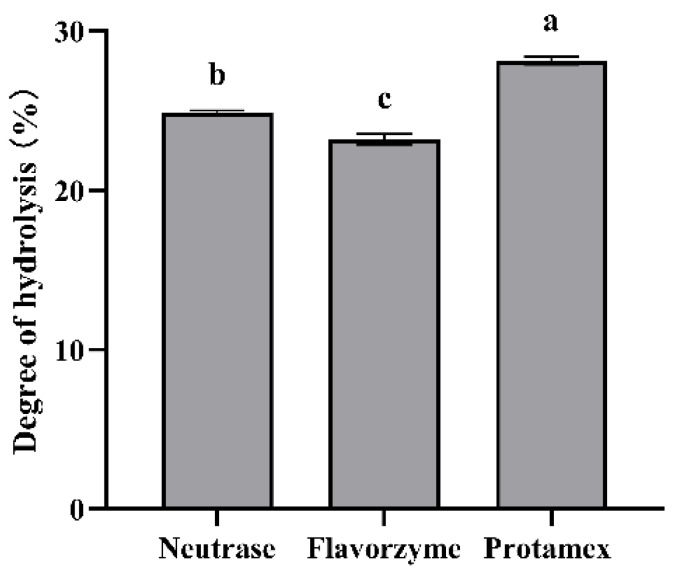
Hydrolysis degree of ESM (*n* = 3, x ± sd). Different letters indicate significant differences (*p* < 0.05).

**Figure 2 foods-13-03828-f002:**
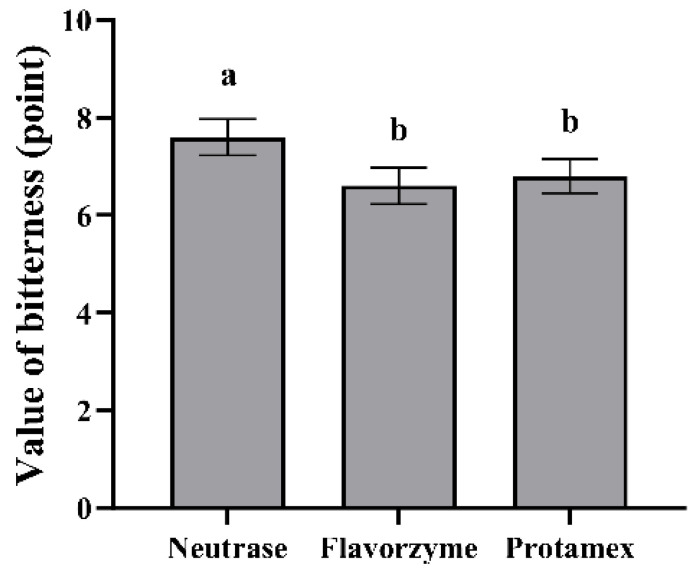
Bitter taste of ESM by three kinds of proteases (*n* = 10, x ± sd). Different lowercase letters indicate significant differences (*p* < 0.05).

**Figure 3 foods-13-03828-f003:**
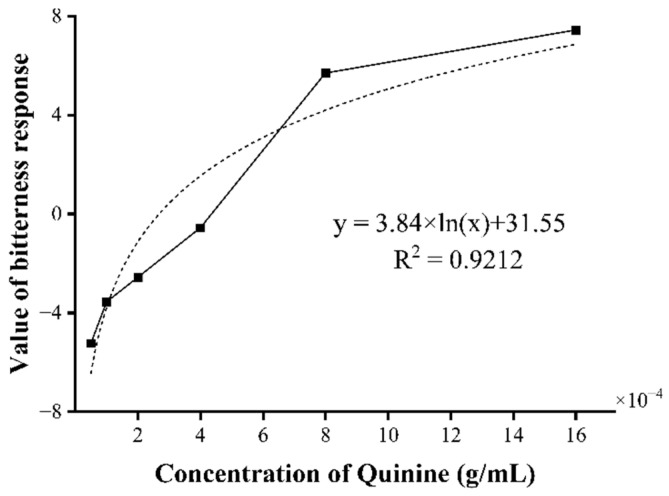
The effect of quinine concentration on bitterness response value.

**Figure 4 foods-13-03828-f004:**
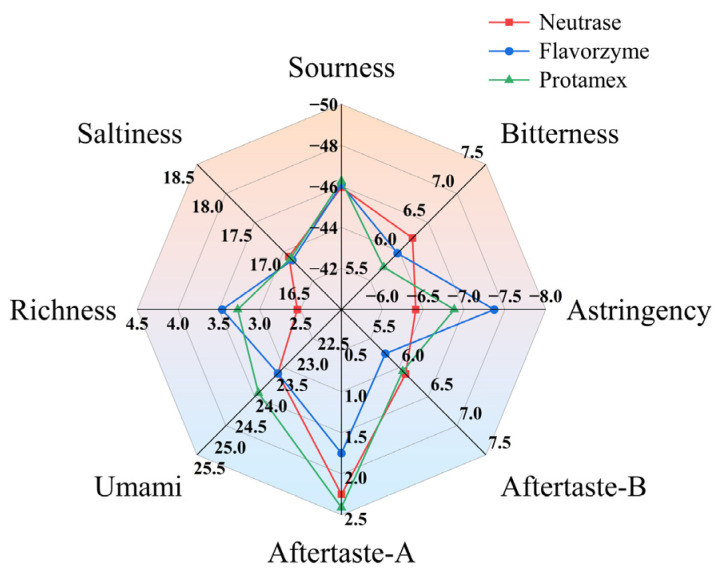
Electronic tongue flavor analysis of ESM.

**Figure 5 foods-13-03828-f005:**
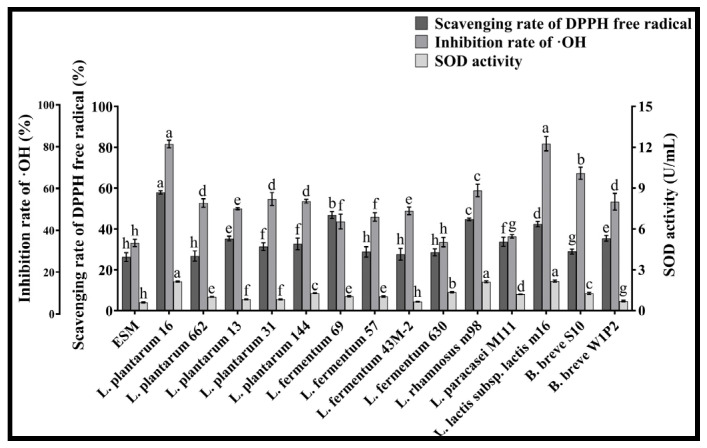
The antioxidant ability of ESM fermented by LAB (*n* = 3, x ± sd). Different lowercase letters of the same indicator indicate significant difference (*p* < 0.05).

**Figure 6 foods-13-03828-f006:**
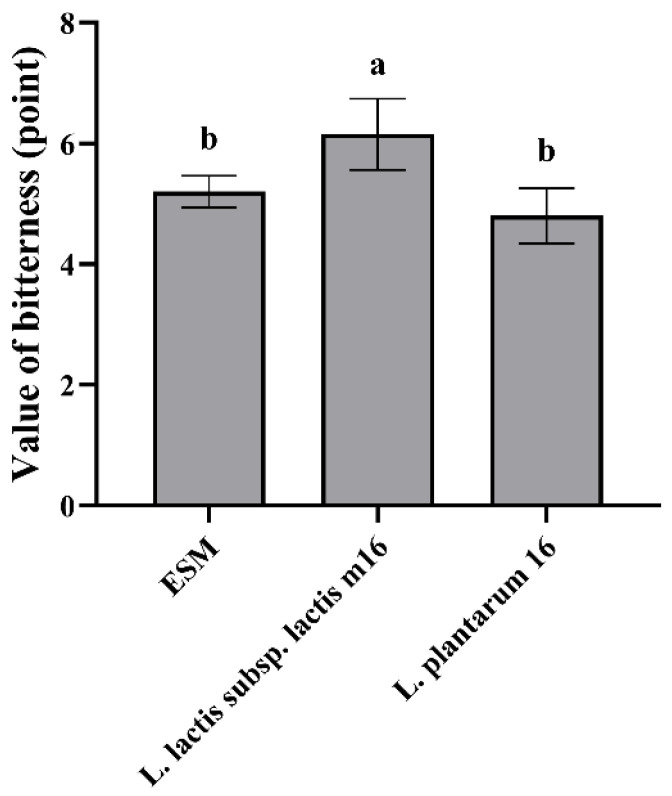
Bitterness evaluation value of ESM fermented by LAB (*n* = 3, x ± sd). Different lowercase letters indicate significant differences (*p* < 0.05).

**Figure 7 foods-13-03828-f007:**
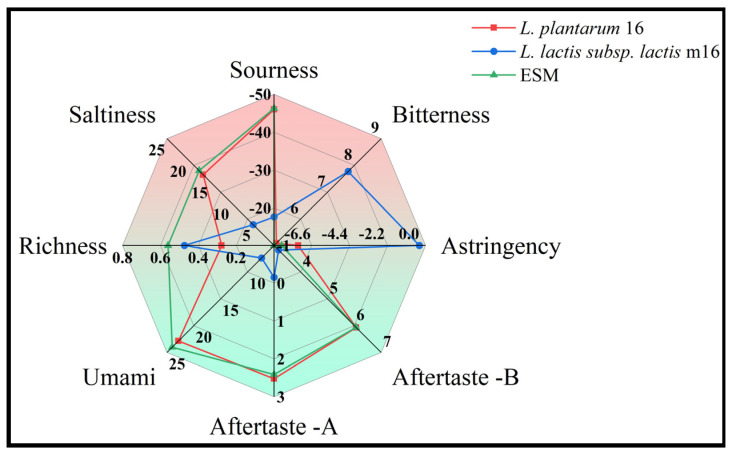
Electronic tongue taste profiles of ESM fermented by LAB.

**Figure 8 foods-13-03828-f008:**
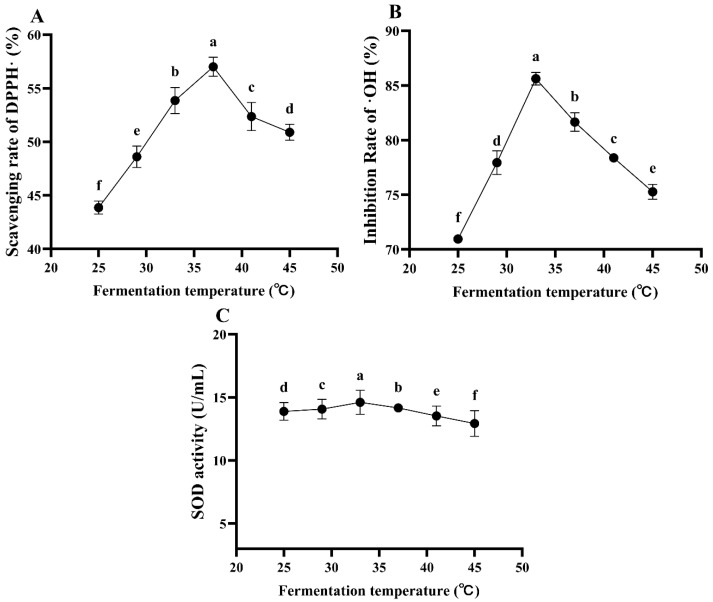
The antioxidant ability of ESM fermented by *L. plantarum* 16 at different fermentation temperatures. (**A**), DPPH free radical scavenging rate; (**B**), ·OH inhibition rate; (**C**), SOD activity. Different lowercase letters indicate significant differences (*p* < 0.05).

**Figure 9 foods-13-03828-f009:**
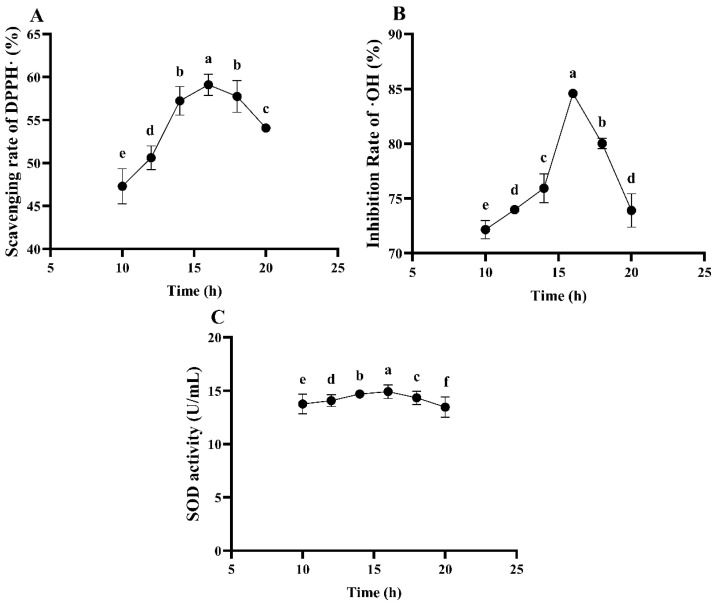
The antioxidant ability of ESM fermented by *L. plantarum* 16 at different fermentation times. (**A**), DPPH free radical scavenging rate; (**B**), ·OH inhibition rate; (**C**), SOD activity. Different lowercase letters indicate significant differences (*p* < 0.05).

**Figure 10 foods-13-03828-f010:**
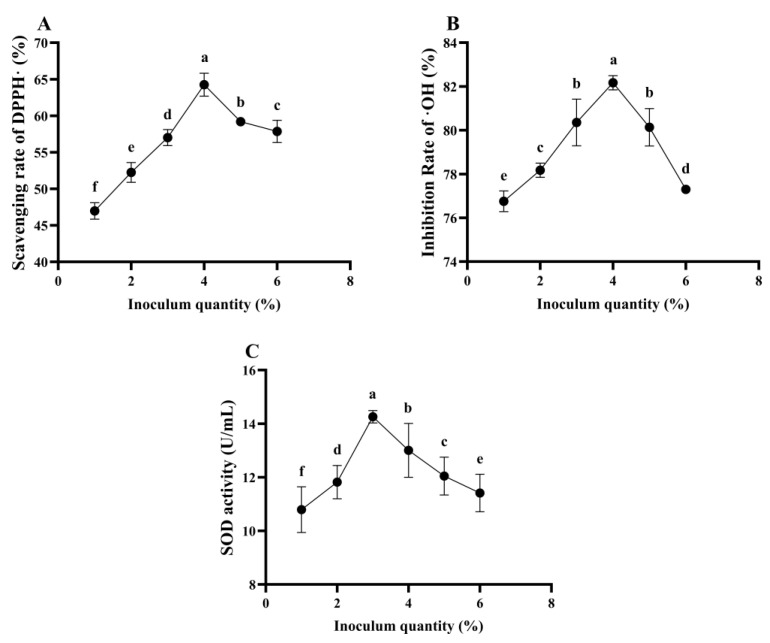
The antioxidant ability of ESM fermented by *L. plantarum* 16 under different inoculum quantity. (**A**), DPPH free radical scavenging rate; (**B**), ·OH inhibition rate; (**C**), SOD activity. Different lowercase letters indicate significant differences (*p* < 0.05).

**Figure 11 foods-13-03828-f011:**
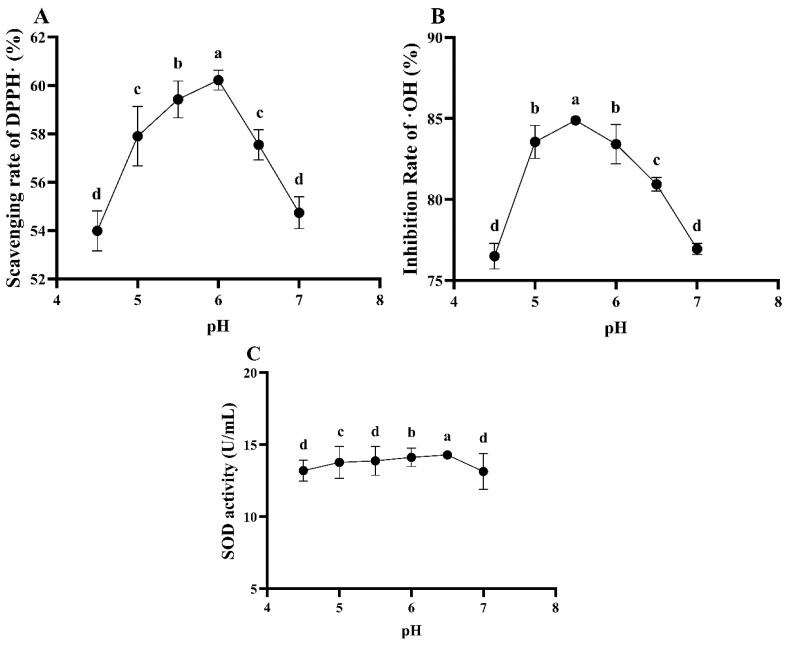
The antioxidant ability of *L. plantarum* 16-fermented ESM under different enzyme release pH conditions. (**A**), DPPH free radical scavenging rate; (**B**), ·OH inhibition rate; (**C**), SOD activity. Different lowercase letters indicate significant differences (*p* < 0.05).

**Figure 12 foods-13-03828-f012:**
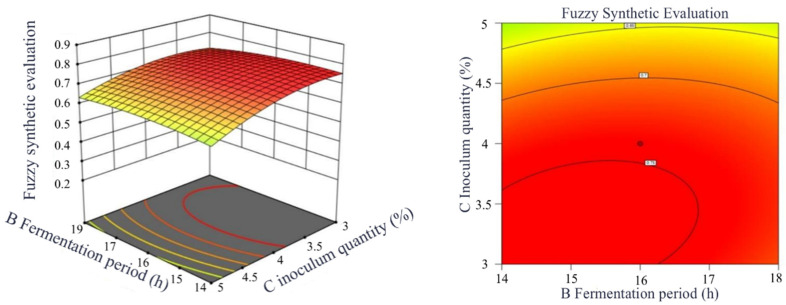
Response surface and contour plots of the impact of interaction on comprehensive evaluation values. The lighter the color, the less the influence of the experimental factors on the fuzzy comprehensive evaluation in this response surface plot.

**Table 1 foods-13-03828-t001:** The optimal conditions for proteases.

Proteinase	Enzymolysis Temperature/(°C)	Enzyme pH	Enzyme Dosage/(U/g)
Neutrase	50	7.0	10,000
Flavorzyme	50	7.0	6000
Protamex	55	7.0	10,000

**Table 2 foods-13-03828-t002:** The level of factor.

Coding	Factor			
	Inoculum Quantity/(%)	Fermentation Temperature/(°C)	Fermentation Time/(h)	ESM pH
−1	3.0	33.0	14.0	5.5
0	4.0	35.0	16.0	6.0
1	5.0	37.0	18.0	6.5

**Table 3 foods-13-03828-t003:** Effect of LAB fermentation on volatile flavor substances in ESM (*n* = 3, x ± sd).

Species	Material	Relative Abundance/(%)		
		Control	*L. Plantarum* 16	*L. lactis* Subsp. *lactis* m16
Aldehydes	Tridecanal			0.15
	Heptanal	0.27		0.09
	Isoamyl aldehyde		0.25	
	Decanal	0.22	0.12	0.34
	Natural nonanal	1.07	0.12	0.35
	Phenylacetaldehyde		0.78	
	2,4-Dimethylbenzaldehyde	1.29		2.96
	Benzaldehyde	5.68		2.59
Subtotal		8.53	1.27	6.48
Alcohols	2-Decanol		0.11	
	1-Octen-3-ol			
	Hexadecanol			
	cis-3-Decanol		0.07	
	(E)-Dec-3-en-1-ol		0.15	
	β-Phenylethanol			
	2-Decen-1-ol		0.09	0.34
	n-Hexanol		0.33	
	3,4-Dimethylbenzyl alcohol		0.18	0.21
	(2S,3S)-(+)-2,3-Butanediol		1.08	
	Undecyl alcohol		0.55	0.51
	Isopentyl alcohol		1.50	
	1-Decanol		1.66	
	1-Heptanol		0.86	0.63
	1-Pentanol			
	Nonyl alcohol		1.90	1.92
	Octanol	1.34	4.52	3.11
Subtotal		1.34	13.00	6.72
Ketones	2-Tridecanone			0.18
	2-Nonanone	0.34	0.17	0.49
	2,3-Heptanedione			
	2-Heptanone	0.30	0.33	0.61
	2-Decanone	1.19	0.23	0.34
	Isophorone	0.26	2.14	3.39
Subtotal		2.09	2.87	5.01
Esters	Ethyl octanoate			
	Methyl 8,11-octadecadienoate	0.25		
	Methyl 12,15-octadecadienoate		0.36	
	Isobutyl 2,2,4-trimethyl-3-hydroxyvalerate			0.35
	Heptyl formate			
	Methyl 7,10-octadecadienoate	0.90		
	Methyl palmitate	2.53	0.14	0.32
	Methyl (Z, Z)-9,12-octadecadienoate		0.77	1.74
	Methyl stearate	4.90		
	2,2,4-Trimethyl-1,3-pentanediol di isobutyrate	5.13	1.54	1.03
	Methyl linoleate	33.90		
Subtotal		47.61	2.81	3.44
Acids	2-Hydroxy-4-methyl pentanoic acid		0.28	
	Caprylic acid			
Subtotal		0.00	0.28	0.00
Heterocyclic compounds	Octane			
	Octadecane			0.47
	Longifolene	0.14	0.10	0.21
	m-Isopropyltoluene			
	Dodecane		0.38	0.21
	2-Undecane		0.13	0.33
	2,4-Di-tert-butylphenol		0.16	0.36
	Azulene	0.19	0.30	
	Tetradecane		0.24	0.47
	1,3,5-Trimethylbenzene			
	Butylated hydroxytoluene	2.68	1.38	2.23
Subtotal		3.01	2.69	4.28

**Table 4 foods-13-03828-t004:** Response surface experimental design scheme and results.

Experiment Number	Influencing Factors				Sensory Assessment	DPPH Free Radical Scavenging Rate/(%)	Inhibition Rate of ·OH/(%)	SOD Activity/(%)	Fuzzy Synthetic Evaluation
	A Fermentation Temperature/(°C)	B Fermentation Time/(h)	C Inoculum Quantity/ (%)	D ESM pH					
1	33	14	4	6	3.75	58.88	48.36	12.22	0.40
2	37	14	4	6	5.00	59.65	46.86	11.52	0.64
3	33	18	4	6	4.75	60.5	77.45	13.32	0.51
4	37	18	4	6	6.00	54.65	53.81	12.17	0.54
5	35	16	3	5.5	7.25	60.01	81.6	12.97	0.44
6	35	16	5	5.5	6.50	59.71	47.66	11.49	0.47
7	35	16	3	6.5	5.00	59.51	52.01	10.50	0.51
8	35	16	5	6.5	5.50	58.59	40.93	11.75	0.45
9	33	16	4	5.5	4.75	59.25	46.75	12.39	0.55
10	37	16	4	5.5	6.75	60.18	39.91	9.23	0.58
11	33	16	4	6.5	3.75	56.05	73.57	12.32	0.51
12	37	16	4	6.5	3.75	56.27	46.08	12.66	0.74
13	35	14	3	6	5.50	59.49	49.01	11.50	0.45
14	35	18	3	6	6.50	59.61	40.87	9.13	0.48
15	35	14	5	6	5.75	54.06	42.57	10.51	0.38
16	35	18	5	6	6.75	58.39	78.76	14.77	0.46
17	33	16	3	6	4.75	55.48	44.08	10.86	0.41
18	37	16	3	6	4.75	56.97	46.43	10.25	0.64
19	33	16	5	6	4.15	52.42	53.53	12.57	0.46
20	37	16	5	6	5.75	62.19	62.68	11.97	0.60
21	35	14	4	5.5	6.25	55.69	44.46	12.18	0.49
22	35	18	4	5.5	7.75	48.5	52.01	10.25	0.50
23	35	14	4	6.5	4.25	59.44	47.43	12.72	0.51
24	35	18	4	6.5	5.50	60.82	38.51	12.44	0.62
25	35	16	4	6	5.75	61.07	44.72	12.27	0.47
26	35	16	4	6	5.50	62.31	41.93	9.96	0.50
27	35	16	4	6	5.25	57.59	43.77	11.38	0.51
28	35	16	4	6	5.25	61.05	85.59	14.04	0.50
29	35	16	4	6	5.50	57.35	43.88	14.17	0.52

**Table 5 foods-13-03828-t005:** Response surface experiment analysis of variance results.

Source of Variance	Sum of Squares	Degree of Freedom	Mean Square	F-Value	*p*-Value	Significant
Model	0.1623	14	0.0116	12.75	<0.0001	Statistical significance
A—Fermentation Temperature	0.0695	1	0.0695	76.44	<0.0001	**
B—Fermentation Duration	0.0045	1	0.0045	4.9	0.044	**
C—Inoculum Quantity	0.0007	1	0.0007	0.7485	0.4015	
D—pH of ESM	0.0083	1	0.0083	9.09	0.0093	**
AB	0.0101	1	0.0101	11.08	0.005	**
AC	0.0023	1	0.0023	2.56	0.1319	
AD	0.0102	1	0.0102	11.23	0.0048	**
BC	0.0008	1	0.0008	0.8772	0.3649	
BD	0.0025	1	0.0025	2.76	0.1189	
CD	0.002	1	0.002	2.21	0.1593	
A^2^	0.0217	1	0.0217	23.88	0.0002	**
B^2^	0.0021	1	0.0021	2.34	0.1483	
C^2^	0.0123	1	0.0123	13.55	0.0025	**
D^2^	0.0061	1	0.0061	6.71	0.0214	**
Residual	0.0127	14	0.0009			
Lack of fit	0.0111	10	0.0011	2.8	0.1665	Insignificant
Pure error	0.0016	4	0.0004			
Sum total	0.1751	28				

** indicates significance at *p* < 0.01.

**Table 6 foods-13-03828-t006:** The impact of *L. plantarum* 16 fermentation on the content of amino acids in ESM.

Amino Acid	ESM (g/100 g)	ESM Fermented by *L. plantarum* 16 (g/100 g)
Aspartic acid	0.270 ± 0.017 ^a^	0.270 ± 0.017 ^a^
Threonine	0.150 ± 0.017 ^a^	0.150 ± 0.017 ^a^
Serine	0.180 ± 0.004 ^a^	0.170 ± 0.005 ^b^
Glutamic acid	0.670 ± 0.010 ^b^	0.690 ± 0.005 ^a^
Proline	0.300 ± 0 ^a^	0.300 ± 0.010 ^a^
Glycine	0.063 ± 0 ^b^	0.064 ± 0.001 ^a^
Alanine	0.104 ± 0.001 ^a^	0.112 ± 0 ^b^
Valine	0.200 ± 0.017 ^a^	0.200 ± 0.017 ^a^
Methionine	0.073 ± 0.001 ^b^	0.077± 0.001 ^a^
Isoleucine	0.165 ± 0.015 ^a^	0.165 ± 0.015 ^a^
Leucine	0.305 ± 0.045 ^a^	0.305 ± 0.045 ^a^
Tyrosine	0.160 ± 0.003 ^a^	0.150 ± 0.005 ^b^
Phenylalanine	0.180 ± 0.004 ^a^	0.160 ± 0.010 ^b^
Histidine	0.104 ± 0.002 ^a^	0.099 ± 0 ^b^
Lysine	0.275 ± 0.005 ^b^	0.286 ± 0.003 ^a^
Arginine	0.115 ± 0.005 ^b^	0.125 ± 0.003 ^a^
Total amount	3.314 ± 0.329 ^a^	3.323 ± 0.135 ^a^

Note: Different letters in the row indicate significant differences (*p* < 0.05).

## Data Availability

The original contributions presented in the study are included in the article, further inquiries can be directed to the corresponding author.

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
