# Peer review of "Lactiplantibacillus plantarum for the Preparation of Fermented Low-Bitter Enzymatic Skim Milk with Antioxidant Ability"

_foods, 2024, doi:10.3390/foods13233828_

Round 1
Reviewer 1 Report
Comments and Suggestions for Authors
The manuscript ensures a completion of the databases regarding the improvement of milk processing techniques through the chosen subject and the obtained results. Increasing the antioxidant potential of food products has an important role for processors and consumers, and the improvement of sensory qualities causes an increase in the degree of acceptability for different categories of consumers. There is the possibility for a more complex graphical representation using the same coordinate system of the comparative results.
Author Response
Comments 1: The manuscript ensures a completion of the databases regarding the improvement of milk processing techniques through the chosen subject and the obtained results. Increasing the antioxidant potential of food products has an important role for processors and consumers, and the improvement of sensory qualities causes an increase in the degree of acceptability for different categories of consumers.
Response 1: It has been revised in line 273.
Reviewer 2 Report
Comments and Suggestions for Authors
This study presents a novel approach by using L. plantarum to reduce both allergenicity and bitterness in hydrolyzed milk, which is unusual compared to similar studies where typically focus on only one of these characteristics. The developed bioprocess to decrease bitternes by converting aldehides into alcohols and other pleasant-tasting compounds is a great idea and may have significant applications in the industry of functional foods. Here are some suggestions to improve the quality and presentation of manuscript.
Abstract. Include the values for the response variables under optimal conditions.
Throughout the manuscript, scientific names appear and are not in italics, please correct. Revise the manuscript to correct spacing details between digits and units.
Line 85: Replace “A 12% (W/W)” with “A 12% (w/w)”.
Table 1: There is some words in Spanish language, please correct.
Citations 11-18 are wrong, put the last name before the citation number “Last name et al. [11]”
Line 99: Replace “..according to the method in the reference [11]” with “according to Guowei et al. [11]”.
Line 105: Replace “..according to the standard scores [12]” with “according to Huang et al. [12]”.
Line 111: Replace “…procedure following the reference method [13]” with “procedure was according to Bruun et al. [13]”.
Line 115: Replace “80% (W/W)” with “80% (w/w)”.
Line 119: Replace “…and the following operations were carried out with the reference method [14]” with “.. and inhibitory activity against ●OH was carried out according to Jing et al. [14]”.
Line 123: Replace “... detailed method described in reference [15]” with “detailed method described by Hucheng et al. [15]”.
Section 2.7 should be prior to section 2.6.2 and 2.6.3. because in them describes analysis of fermented ESM.
Section 2.8. Rewrite this section and cite correctly “Dan et al. [16]” only once.
Line 164. “:.data were analyzed according to the references [17]”. This cite is unnecessary, remove it.
It is recommended to briefly describe the methodologies used, and not only mention the reference.
Line 170: Replace “…and the following operations were carried out the reference method [18]” with “.. and the following operations were carried out to Jaudzeus et al [18]”.
Section 2.11. Include the experimental desing applied, as well as the test of mean comparison.
Line 145-156: This paragraph is redacted in present, please correct it.
Fig. 1,2, 5,6: Include both error bars (top and bottom)
Fig. 3 does not have the same formato as the rest of figures. Please homogenize it.
Figure 8: The axis “x” describe “fermentation tempemture” correct it.
Line 212: The verb is in present form, please change it to past.
Line 232: Replace “… being 81.58% and 81.72%, respectively” with “being 81.58 and 81.72%, respectively”. Apply this change also in lines 274, 277, 279, 280, 299, 316, 333, 334, 335, 336, 337, 348, 350, 457.
Figure 12. Replace “Inoculum dosage” with “inoculum quantity”, according to used terms in table 5.
Table 6. Indicates if the letters indicate significant differences between treatments (ESM and ESM fermented) and add the p-value.
Line 434 and 442: What figure??
Further discussion is needed regarding the effect of each factor evaluated on the response variable to be optimized.
Please check the guide for authors, and use the correct citation style. Put the references according to the guide for authors. You can use a recent published manuscript for a reference.
Author Response
Comments 1: Abstract. Include the values for the response variables under optimal conditions. 32-37
Response 1: It has been revised in line 32-37, and highlight in yellow.
Comments 2: Throughout the manuscript, scientific names appear and are not in italics, please correct. Revise the manuscript to correct spacing details between digits and units. 301
Response 2: It has been revised in line 95-97, and highlight in yellow.
Comments 3: Line 85: Replace “A 12% (W/W)” with “A 12% (w/w)”.
Response 3: It has been revised in line 103, and highlight in yellow.
Comments 4: Table 1: There is some words in Spanish language, please correct.
Response 4: It has been revised in line 110 , and highlight in yellow.
Comments 5: Citations 11-18 are wrong, put the last name before the citation number “Last name et al. [11]”
Response 5: It has been revised in line 113, 117, 124, 144, 157, 174, 208, and highlight in yellow.
Comments 66: Line 99: Replace “..according to the method in the reference [11]” with “according to Guowei et al. [11]”.
Response 6: It has been revised in line 113 and 117, and highlight in yellow.
Comments 7: Line 105: Replace “..according to the standard scores [12]” with “according to Huang et al. [12]”.
Response 7: It has been revised in line 124 , and highlight in yellow.
Comments 8: Line 111: Replace “…procedure following the reference method [13]” with “procedure was according to Bruun et al. [13]”.
Response 8: It has been revised in line 132 , and highlight in yellow.
Comments 9: Line 115: Replace “80% (W/W)” with “80% (w/w)”.
Response 9: The method has been revised in line 140-145, and “80% (W/W)” has been deleted.
Comments 10: Line 119: Replace “…and the following operations were carried out with the reference method [14]” with “.. and inhibitory activity against ●OH was carried out according to Jing et al. [14]”.
Response 10: The method has been revised in line 147-151, and highlight in yellow.
Comments 11: Replace “... detailed method described in reference [15]” with “detailed method described by Hucheng et al. [15]”.
Response 11: It has been revised in line 157, and highlight in yellow.
Comments 12: Section 2.7 should be prior to section 2.6.2 and 2.6.3. because in them describes analysis of fermented ESM.
Response 12: It has been revised in line 134-137, and highlight in yellow.
Comments 13: Section 2.8. Rewrite this section and cite correctly “Dan et al. [16]” only once.
Response 13: It has been revised in line 174, and highlight in yellow.
Comments 14: Line 164. “:.data were analyzed according to the references [17]”. This cite is unnecessary, remove it.
Response 14: It has been deleted in line 202, and highlight in yellow.
Comments 15: It is recommended to briefly describe the methodologies used, and not only mention the reference.
Response 15: It has been revised in line 111-174, and highlight in yellow.
Comments 16: Line 170: Replace “…and the following operations were carried out the reference method [18]” with “.. and the following operations were carried out to Jaudzeus et al [18]”.
Response 16: It has been revised in line 208, and highlight in yellow.
Comments 17: Section 2.11. Include the experimental desing applied, as well as the test of mean comparison.
Response 17: It has been revised in line 210-213, and highlight in yellow.
98 to 103
Comments 18: Line 145-156: This paragraph is redacted in present, please correct it.
Response 18: We did not find traces of editing in lines 1-2, but we found traces of editing in line 141 and made modifications, as shown in line 178, and highlight in yellow.
Comments 19: Fig. 1,2, 5,6: Include both error bars (top and bottom)
Response 19: It has been revised in line 220, 229, 273, and 285.
Comments 20: Fig. 3 does not have the same formato as the rest of figures. Please homogenize it.
Response 20: It has been revised in line 246.
Comments 21: Figure 8: The axis “x” describe “fermentation tempemture” correct it.
Response 21: It has been revised in line 338.
Comments 22: Line 212: The verb is in present form, please change it to past.
Response 22: It has been revised in line 249, and highlight in yellow.
Comments 23: Line 232: Replace “… being 81.58% and 81.72%, respectively” with “being 81.58 and 81.72%, respectively”. Apply this change also in line 274, 277, 279, 280, 299, 316, 333, 334, 335, 336, 337, 348, 350, 457.
Response 23: It has been revised in line 305, 309, 312, 314, 330, 334, 349, 368, 370, 381, 383, 385, and highlight in yellow.
Comments 24: Figure 12. Replace “Inoculum dosage” with “inoculum quantity”, according to used terms in table 5.
Response 24: It has been revised in line 412 and 422, and highlight in yellow.
Comments 25: Table 6. Indicates if the letters indicate significant differences between treatments (ESM and ESM fermented) and add the p-value.
Response 25: It has been revised in line 448, and highlight in yellow.
Comments 26: Line 434 and 442: What figure??
Response 26: It has been revised in line 469 and 476, and highlight in yellow.
Comments 27: Further discussion is needed regarding the effect of each factor evaluated on the response variable to be optimized.
Response 27: It has been revised in line 503-516, and highlight in yellow.
Comments 28: Please check the guide for authors, and use the correct citation style. Put the references according to the guide for authors. You can use a recent published manuscript for a reference.
Response 28: It has been revised in line 557-665, and highlight in yellow.
Reviewer 3 Report
Comments and Suggestions for Authors
The manuscript "Preparation of low-bitter of Lactiplantibacillus plantarum 16
fermented enzymatic skim milk with antioxidant ability" conducted an experiment to study the effect of LAB on the bitterness, taste, and flavor of enzymatic skim milk (ESM). The authors also used response surface methodology to optimize the conditions of LAB fermentation of ESM. I believe their findings can contribute to scientific knowledge in the field. However, the manuscript needs major revisions as it lacks essential information and fails to explain the study's relevance and demonstrates
an understanding of all aspects of the subject
I see the following major issues that should be resolved:
#1 Introduction section is poorly written and needs to be improved. It is not clear what is the novelty of the study since the authors do not provide enough information on what is in the literature. It is not established how are going to have a great impact on the field and It fails to provide more detailed information to support what is being studied.
Why ESM? What other enzymes are used? Is there anything similar reported in the literature?
#2 The experiments are reasonably well done, but in methodology, the authors need to describe, more precisely, how some of the experiments were conducted. Although it is referenced, portions of the technique are important to be detailed.
#3 The Discussion has too many redundant conclusions, and the authors should go deep on the significance of the results. Were the values expected? How does it compare to others? Are there others reported in the literature?
#4 The manuscript needs deep revision to correct grammar and spelling errors. Also, some sentences are confusing and must be rephrased.
Other additional comments/suggestions are highlighted in the attached file. I hope the comments are constructive and help the authors to improve the manuscript.

The manuscript needs deep revision to correct grammar and spelling errors. Also, some sentences are confusing and must be rephrased.
Author Response
Comments 1: #1 Introduction section is poorly written and needs to be improved. It is not clear what is the novelty of the study since the authors do not provide enough information on what is in the literature. It is not established how are going to have a great impact on the field and It fails to provide more detailed information to support what is being studied.
Why ESM? What other enzymes are used? Is there anything similar reported in the literature?
Response 1: (1) Introduction section has been revised in line 58-61, 74-92, and highlight in yellow.
Comments 2: #2 The experiments are reasonably well done, but in methodology, the authors need to describe, more precisely, how some of the experiments were conducted. Although it is referenced, portions of the technique are important to be detailed.
Response 2: It has been revised in line 112-174, and highlight in yellow.
Comments 3: #3 The Discussion has too many redundant conclusions, and the authors should go deep on the significance of the results. Were the values expected? How does it compare to others? Are there others reported in the literature?
Response 3: It has been revised in line 449-461, 503-527, and highlight in yellow.
Comments 4: #4 The manuscript needs deep revision to correct grammar and spelling errors. Also, some sentences are confusing and must be rephrased.
Response 4: We have made every effort to correct grammar, spelling errors and some sentences in this manuscript.
Comments 5: Other additional comments/suggestions are highlighted in the attached file. I hope the comments are constructive and help the authors to improve the manuscript.
Response 5: Some of the responses are as follows, while others are highlight in blue in manuscript.
Comments 6: The title must be corrected. It is confusing. Low-bitter is a characteristic of the ESM, right?
Response 6: The title has been replaced with “Lactiplantibacillus plantarum for the preparation of fermented low-bitter enzymatic skim milk with antioxidant ability”, and highlight in blue in line 2-3.
Comments 7: Why 12%?
Response 7: Research has shown that a culture medium containing 12% skim milk can promote the growth of lactic acid bacteria (LAB) [1], which has been widely used by many researchers to cultivate LAB [2-4].
[1]Kim, S.H.; Lim, C.H.; Lee, C.; An, G. Optimization of Growth and Storage Conditions for Lactic Acid Bacteria in Yogurt and Frozen Yogurt. Journal of the Korean Society for Applied Biological Chemistry 2009, 52, 76-79.
[2] Moneeb, A.H.; Mehany, T.; Abd-Elmonem, M.A.; Tammam, A.A.; Zohri, A.-N.A.; El-Desoki, W.I.; Esatbeyoglu, T. Probiotic Lactobacillus strains as protective adjunct cultures against fungal growth and toxin production in Hard cheese. LWT 2024, 117057
[3] Jingyi, Z.; Tao, W.; Dziugan, P.; Hongfei, Z.; Bolin, Z. Increasing lactose concentration is a strategy to improve the proliferation of Lactobacillus helveticus in milk. Food Science & Nutrition 2020, 9, 1050-1060.
[4] Ye, S.; Yu, T.; Yang, H.; Li, L.; Wang, H.; Xiao, S.; Wang, J. Optimal culture conditions for producing conjugated linoleic acid in skim-milk by co-culture of different Lactobacillus strains. Annals of microbiology 2013, 63, 707-717.
Comments 8: How was the optimal conditions for each protease defined?
Response 8: Commercial enzymes (Novozymes Biotechnology Co., Ltd, Tianjin, China) were used to hydrolyze skim milk in this study; therefore, the hydrolysis conditions were used according to the product instructions and the details are presented in Table 1 of this manuscript.
Comments 9: This umami-tasting acetyl derivatives is desired?
Response 9: Yes, umami-tasting acetyl derivatives could improve the bitterness of foods. A study had found that L. plantarum had the ability to convert bitter amino acids into acetyl derivatives with umami taste[5], which might be the possible reason why the bitterness of ESM fermented by L. plantarum 16 was lower than that of L. lactis subsp. lactic m16, while the umami was higher in this study.
[5] Han, X.; Pengyan, G.; Zhen-Ming, L.; Fang-Zhou, W.; Li-Juan, C.; Jin-Song, S.; Hui-Ling, Z.; Yan, G.; Xiao-Juan, Z.; Zheng-Hong, X. Changes in physicochemical characteristics and metabolites in the fermentation of goji juice by Lactiplantibacillus plantarum. Food Bioscience 2023, 54, 102881.
Comments 10: “the relative content of 2-hydroxy-4-methylpentanoic acid in the ESM fermented by L. plantarum 16 was 0.28% ”is there any explanation as to why this happened?
Response 10: This may be a normal metabolic product of L. plantarum-fermented food. It has been reported that this flavor substance is also found in Mulberry Juice and rosa roxburghii Tratt juice fermented by L. plantarum [6,7].
[6] Guan, X.; Zhao, D.; Yu, T.; Liu, S.; Chen, S.; Huang, J.; Lai, G.; Lin, B.; Huang, J.; Lai, C.; et al. Phytochemical and Flavor Characteristics of Mulberry Juice Fermented with Lactiplantibacillus plantarum BXM2.Foods 2024,13, 2648.
[7] Luo, Y.; Tang, R.L.; Qiu, H.; Song, A.X. Widely targeted metabolomics-based analysis of the impact of L. plantarum and L. paracasei fermentation on rosa roxburghii Tratt juice. International Journal of Food Microbiology 2024, 417,110686.
Comments 11: 2-Hydroxy-4-methylpentanoic acid could can improve inflammation, fibrosis, and metabolic disorders in the organism?
Response 11: Yes, a study published ”Cell Metabolism” suggests that 2-hydroxy-4-
methylvaleric acid could improve liver inflammation and fibrosis by suppressing the intrinsic HIF-2α-ceramide pathway [8].
[8] Zhang, Y.; Wang, X.; Lin, J.; Liu, J.; Wang, K.; Nie, Q.; Ye, C.; Sun, L.; Ma, Y.; Qu, R.; et al. A microbial metabolite inhibits the HIF-2α-ceramide pathway to mediate the beneficial effects of time-restricted feeding on MASH. Cell metabolism 2024, 36, 1823-1838.